# Chitosan and Natural Rubber Latex Biocomposite Prepared by Incorporating Negatively Charged Chitosan Dispersion

**DOI:** 10.3390/molecules25122777

**Published:** 2020-06-16

**Authors:** Siwarote Boonrasri, Pongdhorn Sae–Oui, Pornchai Rachtanapun

**Affiliations:** 1Faculty of Engineering and Agro-Industry, Maejo University, Chiang Mai 50290, Thailand; 2MTEC, National Science and Technology Development Agency (NSTDA), Pathumthani 12120, Thailand; pongdhor@mtec.or.th; 3Division of Packaging Technology, Faculty of Agro-Industry, Chiang Mai University, Chiang Mai 50100, Thailand; 4Cluster of Agro Bio-Circular-Green Industry (Agro BCG), Chiang Mai University, Chiang Mai 50100, Thailand; 5Center of Excellence in Materials Science and Technology, Faculty of Science, Chiang Mai University, Chiang Mai 50200, Thailand

**Keywords:** natural rubber latex, chitosan, biocomposite, composite, bacteria

## Abstract

Generally, natural rubber/chitosan (NR/CT) biocomposites could be prepared by either mixing natural rubber latex (NRL) with CT acid solution or mixing dry NR with CT powder on mixing equipment. In the present work, a new mixing method has been proposed and properties of the obtained NR/CT biocomposites are investigated. CT particles were prepared to have a negative charge that could be dispersed in water by using a ball mill before mixing with NRL. The effects of CT loading varied from 0 to 8 phr on latex properties and physical properties of NR/CT biocomposite films were focused. The results showed that the viscosity of NRL increased with increasing CT loading. With increasing CT loading from 0 to 8 phr, 300% modulus of the NR/CT biocomposite film increased, whereas the opposite trend was found for elongation at break. Additionally, the presence of CT in the biocomposite resulted in an increased elastic modulus (E’) in conjunction with enhanced antibacterial activity against *Staphylococcus aureus* (*S. aureus*).

## 1. Introduction

Natural rubber latex (NRL) is a colloid system comprising of rubber particles dispersed in water. The rubber particles are covered by a layer of negatively charged proteins and lipids, which provides colloidal stability to the NRL [1]. NRL undergoes coagulation at room temperature by adding an acid such as formic acid and other acids or salts, i.e., calcium chloride, calcium nitrate, etc. [2]. The products from NRL such as tubing, pillow foam, rubber nipple, surgical glove, and catheter find extensive applications in the biomedical field [3]. The advantages of these products are their elasticity and flexibility [4]. However, the products from NRL show some weaknesses such as low modulus, barrier properties and electrical properties that affect the quality of the natural rubber (NR) products. Consequently, many attempts have been made to use a reinforcing filler such as graphene oxide [5], multiwalled carbon nanotube [6], organoclay [7,8], carbon black [9], silica [10], cassava starch [11], and carbon fiber [12], to improve NR properties. The incorporation of bio-fillers from renewable resources into NRL could result in new biocomposites with enhanced features and biodegradability, so it is of interest to study the possibility of using chitosan (CT) as a reinforcing filler for NRL.

CT is a linear copolymer composed of β-(1–4)-linked N-acetyl-d-glucosamine obtained by partial chitin deacetylation typically found in the form of granules, sheets, or powders. There are currently a variety of chitosan applications in the biomedical, food, and chemical industries owing to its attractive properties such as biodegradability, biocompatibility, low toxicity, biological activity, and excellent potential as an antimicrobial agent [13,14,15].

In previous works, CT was used to blend with natural rubber (NR) or epoxidized natural rubber (ENR) to improve thermal resistance [16], mechanical properties [17], and electric properties [18] of rubber compounds. Several approaches have been reported for the preparation of NR/CT or ENR/CT biocomposites, including the solution mixing in which CT is dissolved in acid solution before being combined with NRL [19,20,21,22,23] and the dry mixing in which CT is added directly into dry rubber on a 2-roll mill [24]. In the former, CT is entirely dissolved in distilled water containing 2% (v/v) acetic acid before being added into a mixture of NRL and curing agents. This method may cause poor CT dispersion because rubber particle surfaces have negative charges and, when CT acid solution is added, the rubber particles are quickly coagulated. To alleviate the problem, the alternative mixing method is proposed. In this work, CT powder was predispersed in water without acid. The CT particles were prepared to have a negative charge that could be dispersed in water by using a ball mill. In view of this, we have focused on incorporating a new type of CT (dispersion form) into NRL compounds. The effect of CT loading on latex properties, tensile properties, morphology, and antimicrobial efficacy of NR/CT biocomposites was reported.

## 2. Materials and Methods

### 2.1. Materials

High-ammonia (HA type, 0.7% NH_3_) concentrated NRL with 60% of dry rubber content (DRC) was manufactured by Thai Rubber Latex (Samutprakan, Thailand). Chitosan (CT) powder produced from crab shells having a particle size of 100 mesh (149 μm) was supplied by Taming-enterprises (Samutsakhon, Thailand). Other ingredients including 10% potassium hydroxide (KOH) solution, 10% potassium oleate solution, 50% sulfur (S) dispersion, 50% zinc diethyldithiocarbamate (ZDEC) dispersion, 50% zinc oxide (ZnO) dispersion, 50% phenolic antioxidant (PA) dispersion, bentonite clay, sodium naphthalene sulfonate, and ammonia (NH_3_) were all purchased from Lucky Four Co., Ltd. (Nonthaburi, Thailand).

### 2.2. Preparation and Characterization of 10% CT Dispersion

Porcelain ball mill, with a ball diameter of 1 cm, was used to prepare the 10% CT dispersion in this work. Table 1 shows the compositions used in the preparation. All ingredients were mixed and ground using a ball mill at a milling speed of 40 rpm for four days to get sufficiently small particle size of CT. The pH and viscosity of the 10% CT dispersion were then measured. The final particle size of the CT was determined using an optical microscope (CX23, Olympus, Japan) with a magnification of 40×.

### 2.3. Preparation of NRL/CT Biocomposites

The 10% CT dispersion was mixed with NRL and other ingredients according to the formulation given in Table 2. The amount of CT was varied from 0 to 8 phr. Water was added into the latex compound for dilution to a total solid content of 45%. The mixture was then heated at 45 °C and stirred in a mixing vessel for 24 h and finally cooled down at room temperature for 24 h before being tested.

The total solid content (TSC, %) of the latex compound was determined according to ISO 124:2014. Alkalinity, %NH_3_, was measured conforming to ISO 125:2011. pH of the latex compound was measured by using a pH meter, conforming to ISO 976:2013. Potassium hydroxide number (KOH No.), the weight in grams of potassium hydroxide reacting with ammonium ion of the latex compound was determined according to ISO 127:2018. Viscosity of the latex compound was measured by using a Brookfield DV III ultra (Brookfield, USA), conforming to ISO 1652:2004. All viscosity values are expressed in centipoise (cps).

The chloroform number (CN) test was performed through the coagulation of an NRL via mixing with an equal volume of chloroform. After 2–3 min, the coagulum was examined and graded according to the texture of the coagulum. The CN was expressed as follows: (1) unvulcanized, (2) lightly vulcanized, (3) moderately vulcanized, and (4) fully vulcanized.

### 2.4. Preparation of NR/CT Biocomposite Films

NR/CT biocomposite films were prepared by the dipping process. Cleaned glass tubes were dipped into a 20% calcium chloride solution for 10 s and dried in a hot air oven at 100 °C for 5 min. Then, the glass tubes were dipped into the latex compound for 1 min, followed by drying in the oven (UN110, Memmert, Germany) at 110 °C for 40 min. The dried NR/CT biocomposites were taken out from the glass tubes, cut along the length of the glass tube to form composite films and conditioned at room temperature (25 °C) with relative humidity of 65% for 24 h before being tested for their physical and dynamic properties.

### 2.5. Characterization of Physical Properties

Dumb-bell shaped test pieces were cut from the NR/CT biocomposite films. The tensile test was done following ISO 37:2017 using a computerized tensile tester (Instron 5565, Massachusetts, United States) with a load cell of 10 kN. The crosshead speed of the Instron machine was set at 500 mm/min. Modulus at 300% elongation (300% modulus) and elongation at break (EB) were analyzed from the tensile data. The values of tensile properties were the average of 5 measurements. The mechanical properties of aged samples were studied by aging the specimens in the hot air oven at 100 °C for 22 h. Then, the specimens were cooled down at room temperature for at least 16 h before testing.

Damping property (tan δ) and elastic modulus (E’) were measured as a function of temperature by using the dynamic mechanical thermal analyzer, DMTA (Explexor TM 25 N, Gabo Qualimeter, Germany). The test was evaluated in tension mode at a constant frequency of 10 Hz, static strain of 1%, and a dynamic strain of 0.1%. The temperature was scanned from −80 to −40 °C with a heating rate of 2 °C/min.

Morphology of the NRL/CT biocomposites was studied by using a field emission scanning electron microscope (FE–SEM, model S–4700, Hitachi, Japan) at 5 kV electron energy. The newly cryogenic fractured surfaces of the NR/CT biocomposites were coated with Pt–Pd prior to being examined.

### 2.6. Evaluation of Anti-Microbial Activity

The anti-microbial study was carried out by using an agar diffusion method [25]. The NR/CT biocomposite films were cut with the dimensions of 10 mm × 10 mm and placed on a Mueller Hinton agar medium (Merck, Germany), which had been previously seeded with 105 cfu/mL (Colony Forming Units/mL) of inoculums containing the tested bacterium (*S. aureus*). The plates were then incubated at 37 °C for 24 h. The inhibitory effect on microbial growth expressed the antimicrobial properties of the NR/CT biocomposite films based on the growth inhibition zone diameter (in cm) or clear zone diameter around the NR/CT biocomposite films.

## 3. Results and Discussion

### 3.1. Characteristics of the 10% CT Dispersion and Morphology of the NR/CT Biocomposites

The CT used in this experiment was in the form of fine powder and soluble in acetic acid. However, we did not dissolve CT powder in acid solution because it would cause coagulation when added into NRL. Therefore, CT powder was transformed into a dispersion form without acid before being added into the latex. It was found that after the addition of 10% CT dispersion, the rubber particles in the latex did not coagulate.

The pH of the 10% CT dispersion was 10.26, which was close to that of the latex (Table 3); therefore, 10% CT dispersion could be incorporated into NR latex without coagulation. Theoretically, CT has a positive charge, but it may be changed into a negative charge at high pH value due to the presence of the alkaline substance and sodium naphthalene sulfonate in the formula (Table 1). From the viscosity test, it was found that the CT dispersion had relatively high viscosity (3000 cps).

Figure 1 depicts microscopic images of CT in different forms. The particle size of commercial CT was considerable (ranging from 50 to 100 μm, as shown in Figure 1a, highlighted with a blue circle). However, after grinding by the ball mill technique, the particle size of the CT was significantly reduced (ranging from 1 to 5 μm, as shown in Figure 1b, highlighted with a blue circle). From Figure 1, this indicates that CT particles could be broken down into smaller particles during the ball mill mixing and then sodium naphthalene sulfonate (SNS) acting as a dispersing agent covers the CT particles, as shown in Figure 2b. Therefore, the CT particles were stable because SNS shows a negative charge around the CT particles.

Figure 3a,b show the SEM micrographs taken at a magnification of 20k of the NR/CT biocomposites containing 4 phr and 8 phr of CT. The white spots and high ridges represented the CT particles (indicated by arrows) embedded in the NR matrix (a grey area). It could be perceived that the particle size of CT particles was relatively small, falling in the range of 1–2 μm. It was clear from the micrographs that a fine and uniform phase distribution was exhibited in the NR/CT biocomposites. Figure 3c,d show the SEM micrographs taken at a magnification of 40k of the NR/CT biocomposites containing 4 phr and 8 phr of CT. The CT particles could be seen more clearly at higher magnifications. From Figure 3d, there was no formation of holes or voids at the interface between rubber and CT indicating the good rubber–CT interaction in the NR matrix. Again, SNS can act as a dispersing agent and thus improve the interaction between rubber chains and CT particles, allowing a good dispersion of CT in the rubber matrix. In Figure 2b, the incorporation of SNS introduces the formation of hydrogen bonds between sulfonate groups of SNS and the hydroxyl and amine groups on the CT surface. Additionally, chemical reactions between the sulfonate groups with the hydroxyl and amine groups and rubber molecules may occur during the compound mixing and these should enhance filler–rubber interactions resulting in a better CT dispersion in the rubber matrix. 

### 3.2. NRL/CT Latex Properties

It could be seen from Table 3 that %TSC, pH, KOH No., and CN were not significantly changed with increasing CT loading. Again, the pH of the 10% CT dispersion was 10.26, which was close to that of the latex; therefore, 10% CT dispersion could be incorporated into NR latex without coagulation resulting in a good CT dispersion in NRL matrix as shown in Figure 4. The results also showed that the latex compounds contain slightly higher %NH_3_ compared to the NRL. This occurred because of the existence of an alkaline substance in the CT dispersion.

From Figure 5, it could be seen that the viscosity of the latex compounds increased with an increase of CT loading. This could be described by the high viscosity of the CT dispersion as previously mentioned. Generally, the viscosity suitable for latex processing should be below 200 cps. It could be seen that the latex compounds containing CT of 0.5–4.0 phr had relatively low viscosity and thus could be easily processed. However, the mixture with 8 phr of CT had very high viscosity and thus it was difficult to be processed. Nevertheless, the viscosity of the latex compound could be reduced by diluting with water.

### 3.3. Physical Properties of NR/CT Biocomposites

As expected, the elongation at break (Figure 6a) decreased with increasing CT loading because the CT could reduce the molecular mobility of rubber chains. In addition, the dilution effect may be used to explain the results because CT is much less extendable compared to NR.

The reduction in the elongation at break was found simultaneously with the increase of stiffness or modulus of the NR/CT biocomposites. Again, the increase of the 300% modulus with increasing loading of CT (Figure 6b) was thought to arise from the dilution effect because CT is much stiffer than NR. Other researchers also reported that the modulus value increases when the filler loading is increased [26]. This result highlights the fact that the surface activity of chitosan determines the rubber–filler interaction, agglomeration, and filler particle dispersion hence control the modulus of natural rubber composites. As previously mentioned, the SNS improved CT dispersion in the rubber matrix, giving rise to the increase of modulus with increasing loading of CT. Similar observation has been reported by Berki et al., who studied the addition of graphene oxide into natural rubber through latex precompounding [5]. It is found that the property improvements were attributed to a better dispersion of the graphene oxide after latex precompounding compared to the nanocomposites produced via direct melt mixing of the natural block rubber.

Surprisingly, CT prepared in a dispersion form brought more considerable reinforcement, resulting in higher modulus, than other preparation methods, including CT acid solution mixed with NRL [23], and CT dry-mixed by a 2-roll milling process [24] as shown in Table 4.

The changes in mechanical properties with hot air aging, viz. 300% modulus and elongation at break, are shown in Figure 6 and Figure 7. It could be seen that elongation at break and the 300% modulus of all NR/CT biocomposites decreased after being exposed to hot air aging. The oxidation effect could be used to explain the results, and chitosan could not help to prevent the oxidation of rubber.

The dynamic mechanical analysis was a useful tool for obtaining some indirect evidence on the dispersion and rubber–graphene interaction [5,27]. From dynamic mechanical measurements, the elastic modulus and tan δ for the NR/CT biocomposites were obtained and plotted against the temperature, as shown in Figure 7a,b, respectively. At the transition zone, the elastic modulus significantly decreased with increasing temperature due to the higher molecular movement ability. It could be observed that, at the temperature below the glass transition temperature (T_g_), modulus of the biocomposites increased with increasing CT loading mainly due to a higher stiffness or modulus of the glassy CT.

It is clearly seen from Figure 8b that the tan δ peak and tan δ area tended to decrease with increasing CT loading. Since the tan δ area represents the amount of rubber molecule participating in the transition, the results indicated that the increase of CT loading led to the reduction of rubber participating in the transition, which could be easily explained by the dilution effect.

It is widely accepted that the temperature at which the tan δ value reaches its maximum is the glass transition temperature (Tg) [10]. The biocomposite without the presence of CT had Tg of −50 °C, and the biocomposite containing CT of 8 phr had T_g_ of −48 °C. The slight increase of Tg in the presence of CT might arise from the interaction between NR and CT, making the movement of rubber molecules more difficult.

### 3.4. The Bacterial Growth Inhibition of NR/CT Films

The antibacterial activity of the NR/CT biocomposite films, represented by the outer diameter of the clear zone (free-living microorganism), is displayed in Figure 8a–c. It was found that all NR/CT biocomposite films showed clear halo zones indicating that these NR/CT biocomposite films could inhibit the growth of *S. aureus* bacteria. Surprisingly, the unfilled composite also offered antibacterial activity against *S. aureus*. It has previously been reported that the NR latex glove has a superior barrier against bacterial transmigration than those made from nitrile and vinyl gloves [28,29,30]. Besides, it is also reported that NR latex gloves give superior water leakage and viral penetration to vinyl gloves [31,32]. It is thought that the anti-bacterial activity found in the unfilled composite arises from the presence of some chemicals, which could act as antimicrobial agents such as S [33], ZnO [34], and ZDEC [35,36].

The antimicrobial efficacy of the NR/CT biocomposite was slightly enhanced when CT was added. Results reveal that rubber film containing CT is useful for the fabrication of rubber catheters requiring good antibacterial activity against the growth of *S. aureus* [37]. Moreover, it is found that the chitosan exhibits outstanding antimicrobial activities for developing antimicrobial packaging films [38]. It has been stated that the antimicrobial activity of chitosan results from its cationic nature [39].

## 4. Conclusions

CT powder could be prepared in the form of dispersion before being added to NRL. The viscosity of the latex compound increased with increasing CT loading. With increasing CT loading from 0 to 8 phr, 300% modulus of the NR/CT biocomposite film was significantly increased whereas the opposite trend was found for the elongation at break. Additionally, the presence of CT exhibited lower tan δ_max_ and higher T_g_. The unfilled composite could provide a certain level of antibacterial activity when tested using *S. aureus*. The antimicrobial efficacy of the NR film was further enhanced with the addition of CT. Results obtained highlight the potential use of this NR/CT biocomposite in the production of antibacterial gloves and other latex products that require high modulus and antibacterial activity such as tubing, pillow foam, a rubber nipple, and a catheter.

## Figures and Tables

**Figure 1 molecules-25-02777-f001:**
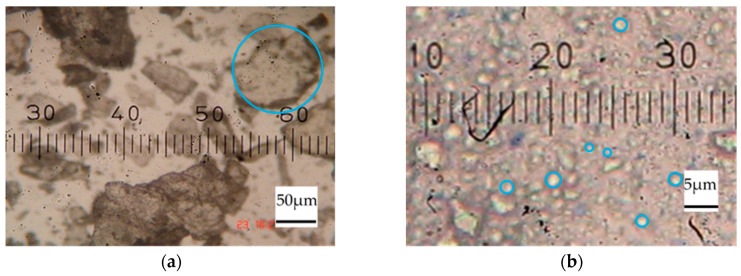
Microscopic images of (**a**) commercial CT and the CT in (**b**) the CT in 10% CT dispersion.

**Figure 2 molecules-25-02777-f002:**
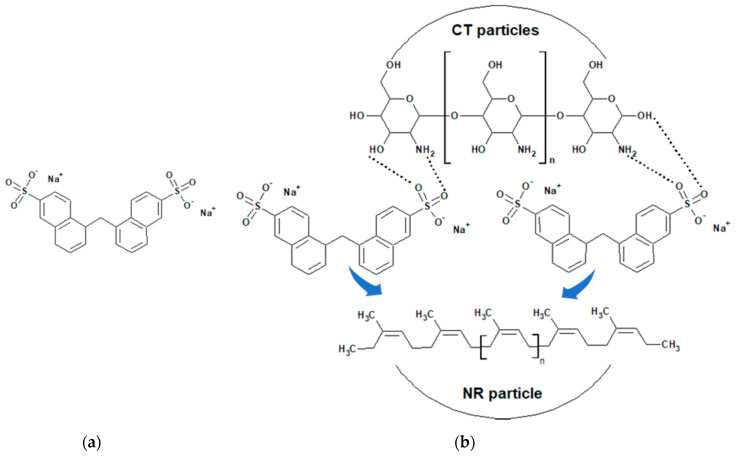
Chemical structure of SNS (**a**) and a model of SNS-coated natural rubber (NR) particle (**b**).

**Figure 3 molecules-25-02777-f003:**
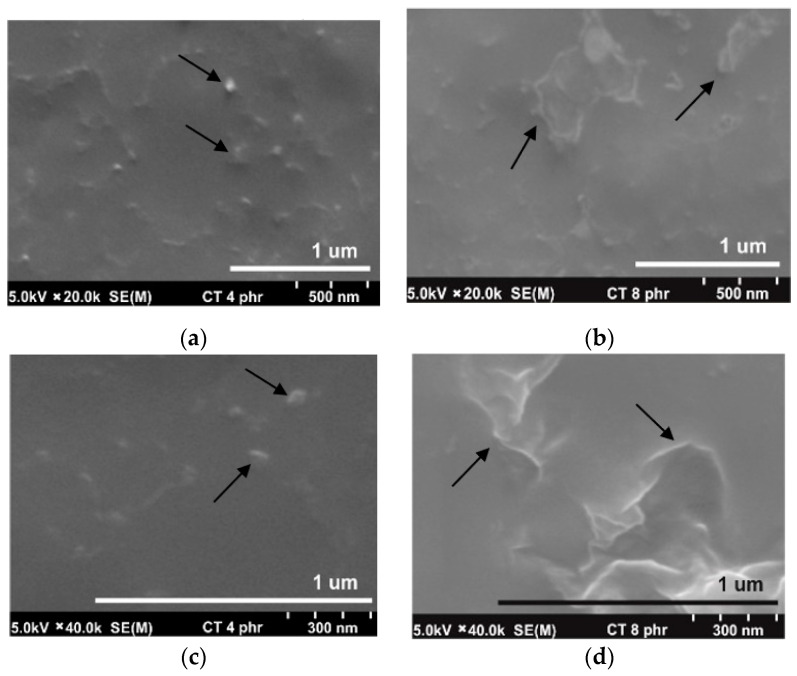
SEM micrograph at magnification of 20k of NRL/CT biocomposites containing CT of 4.0 phr (**a**) and 8.0 phr (**b**) and at magnification of 40k of NRL/CT biocomposites containing CT of 4.0 phr (**c**) and 8.0 phr (**d**).

**Figure 4 molecules-25-02777-f004:**
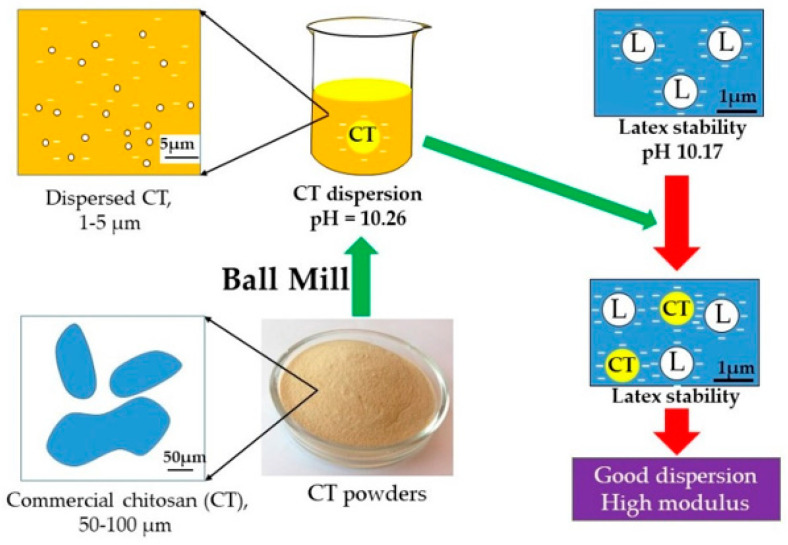
Model of preparation of chitosan dispersion in the natural rubber latex/chitosan compound.

**Figure 5 molecules-25-02777-f005:**
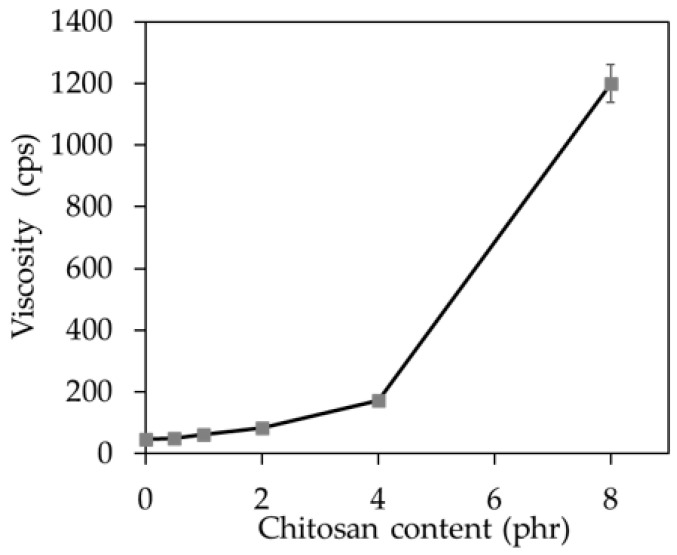
Viscosity of the latex compound with CT loading from 0 to 8 phr.

**Figure 6 molecules-25-02777-f006:**
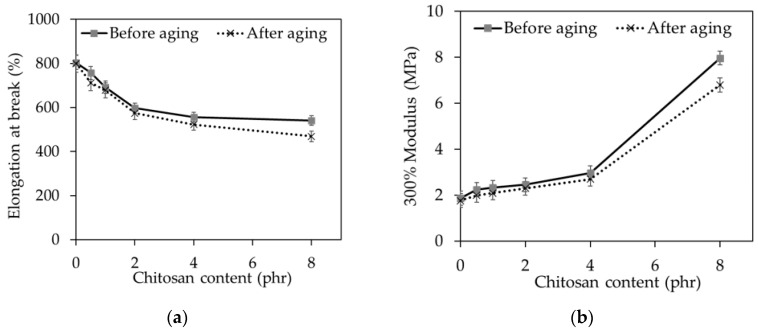
(**a**) Elongation at break and (**b**) the 300% modulus of the NR/CT biocomposites.

**Figure 7 molecules-25-02777-f007:**
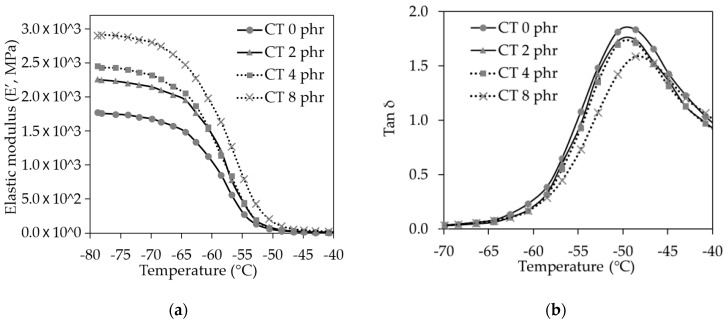
(**a**) Elastic modulus (E’) and (**b**) Tan δ of the NR/CT biocomposites with CT loading from 0 to 8 phr.

**Figure 8 molecules-25-02777-f008:**
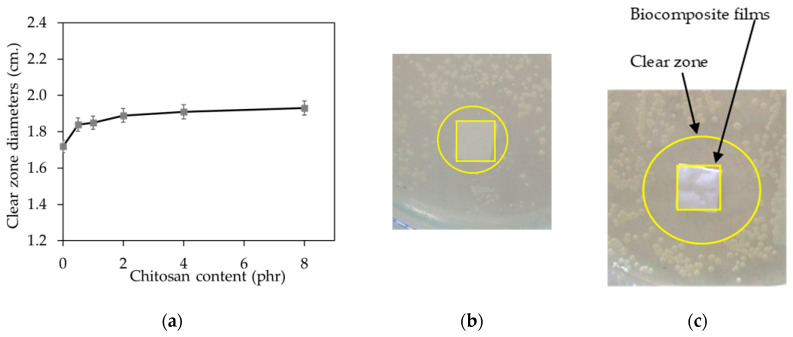
Antibacterial activity of all NR/CT biocomposite films (**a**), picture of unfilled biocomposite (**b**) and picture of biocomposite containing 8 phr of CT (**c**).

**Table 1 molecules-25-02777-t001:** Ingredients for the preparation of 10% chitosan (CT) dispersion.

Ingredient	% (w/w)
Chitosan	10.0
Bentonite clay	1.0
Sodium naphthalene sulfonate	1.0
Water	78.0
NH_3_	10.0
Total	100.0

**Table 2 molecules-25-02777-t002:** Ingredients for the preparation of CT/natural rubber latex (NRL) compounds.

Materials	Loading (phr)					
	1	2	3	4	5	6
60% NRL (HA type)	100.0	100.0	100.0	100.0	100.0	100.0
10% KOH	0.2	0.2	0.2	0.2	0.2	0.2
10% Potassium oleate	0.5	0.5	0.5	0.5	0.5	0.5
50% S	1.0	1.0	1.0	1.0	1.0	1.0
50% ZDEC	0.8	0.8	0.8	0.8	0.8	0.8
50% PA	1.0	1.0	1.0	1.0	1.0	1.0
50% ZnO	2.0	2.0	2.0	2.0	2.0	2.0
10% Chitosan	0.0	0.5	1.0	2.0	4.0	8.0

**Table 3 molecules-25-02777-t003:** Properties of the latex compound with CT loading from 0 to 8 phr.

CT Loading (phr)	% TSC	% NH_3_	pH	KOH No.	CN
0.0	45.67	0.42	10.17	0.65	2
0.5	45.09	0.52	10.23	0.64	2
1.0	45.32	0.52	10.14	0.64	2
2.0	45.82	0.57	10.33	0.62	2
4.0	45.21	0.70	10.45	0.55	2
8.0	45.83	0.96	10.16	0.67	2

**Table 4 molecules-25-02777-t004:** Effect of the different CT preparation methods on the 300% modulus of the NR/CT biocomposite.

Methods	CT Content (phr)	300% Modulus (MPa)
Acid solution mixed	10.0	3.0
Dry mixed	10.0	2.7
Dispersion	8.0	8.1

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
