# Peer review of "Chitosan and Natural Rubber Latex Biocomposite Prepared by Incorporating Negatively Charged Chitosan Dispersion"

_molecules, 2020, doi:10.3390/molecules25122777_

Round 1

Reviewer 1 Report

In this manuscript, the authors present the realization of a chitosan (CT) and natural rubber latex (NRL) composite by incorporating negatively charged CT dispersed in water by using a ball milling to be embedded into the NRL for the compounds. The physical and mechanical properties of the CT/NRL compound were investigated as a function of the CT loading, and also its antimicrobial efficacy of NR/CT biocomposites is investigated.

The paper is well structured, written in comprehensible English, the research is described in detail and well documented. However, some minor revisions and suggestions are recommended before its publication on Molecules. 

Revisions:

  1. The authors should remove excessive adjectives such as “novel method” from the title.

  1. Figure 1: for clarity, the authors should highlight with a circle or a color the different CT particles.

  1. Figure 2: The authors should provide better resolved SEM micrographs of NR/CT biocomposites, the morphology id hardly visible in the presented images.

  1. Page 9: The authors report the antibacterial activity of the NR/CT biocomposite films, represented by the outer diameter of the clear zone in Figure 8, could the authors provide two pictures presenting a comparison with unfilled composite for clarity?

Author Response

Response to Reviewer 1 Comments

Point 1: The authors should remove excessive adjectives such as “novel method” from the title.

Response 1:  We agree that the authors should remove “novel method” from the title and change the title from “A novel method for prepared natural rubber/chitosan biocomposites” to “Chitosan and natural rubber latex  biocomposite prepared by incorporating negatively charged chitosan dispersion”. (with highlight in red)

Point 2: Figure 1: for clarity, the authors should highlight with a circle or a color the different CT particles.

Response 2: Correction has been made in the revised manuscript. (Figure 1, the particle is highlighted with a blue circle)

Point 3: Figure 3: The authors should provide better resolved SEM micrographs of NR/CT biocomposites, the morphology id hardly visible in the presented images.

Response 3: Correction has been made in the revised manuscript. (Figure 3, two more pictures with greater magnification are added.)

Point 4:  Page 9: The authors report the antibacterial activity of the NR/CT biocomposite films, represented by the outer diameter of the clear zone in Figure 8, could the authors provide two pictures presenting a comparison with unfilled composite for clarity?

Response 4: The pictures of clear zone of the biocomposite films have been incorporated in the revised manuscript. (Page 9, Figure 8)

Reviewer 2 Report

The manuscript A novel method for prepared natural rubber/chitosan biocomposites has been structured and written well by following the main steps of scientific work and will be of interest to researchers working in material science. This work deserves to be considered for publication. However, some minor revisions are necessary for publication. To improve the quality of the work, please consider the following suggests:             

Line 41: Why “Organoclay” is written with a capital letter?
Line 42: Why do the new pages start with a black line?
Line 75: Please precisely defined grinding media: type (e.g. ZrO2?), balls diameter and milling speed (rpm). Why did you decide to mix and grind all the ingredients for four days? (and not 3 or 5?)
Line 78:
Description of the microscope CX23:  (Olympus, Tokyo, Japan).
Line 79:
What was the purpose of using bentonite clay to prepare CT dispersion?
Line 88 and  etc. : Please give full ISO name, e.g.: ISO 124:2004, ISO 125:2020, ISO 976: 2013 etc.
Line 93: Correct description: Brookfield DV II ultra: (Brookfield, Middleboro, MA, USA).
Line 105: Please correct that the  UN110 Memmert was produced by a German company.
Line 107: The composite films were conditioned at room temperature but at what RH?
Line 110: According to ISO 37: 2017 dumb-bell spell with hyphens.
Line 118: Correct - tan δ.
Line 150: “Figure 1” – missing space.
Line 229:
“…significantly increased…” Did you make any statistical analysis of the obtained results?
Line 271/ 368: Figure 6 and Figure 8. Please consider adding the 0,5 and 1,0 value (point) at the x-axis, same time remove 6.
Line 365: Correct - tan δmax.
References: Commas, not semicolons, in between the authors' names.

Author Response

Response to Reviewer 2 Comments

Point 1: Line 41: Why “Organoclay” is written with a capital letter?

Response 1: Correction has been made in the revised manuscript. (line 42 with highlight in red)

Point 2: Line 42: Why do the new pages start with a black line?

Response 2: Correction has been made in the revised manuscript. (no black line)

Point 3: Line 75: Please precisely defined grinding media: type (e.g. ZrO2?), balls diameter and milling speed (rpm). Why did you decide to mix and grind all the ingredients for four days? (and not 3 or 5?)

Response 3: Correction has been made in the revised manuscript. (line 76-79  with highlight in red)  Our previous work has shown that a period of 4 days is sufficient to get the required size of CT. Longer period may also be used but relatively small improvement in CT size is obtained.   

Point 4: Line 78: Description of the microscope CX23:  (Olympus, Tokyo, Japan).

Response 4: Correction has been made in the revised manuscript. (line 80-81 with highlight in red)

Point 5: Line 79: What was the purpose of using bentonite clay to prepare CT dispersion?

Response 5: Bentonite clay was used to prevent CT precipitation.

Point 6: Line 88 and  etc. : Please give full ISO name, e.g.: ISO 124:2004, ISO 125:2020, ISO 976: 2013 etc.

Response 6: Correction has been made in the revised manuscript. (Line 90 onwards with highlight in red)

Point 7: Line 93: Correct description: Brookfield DV II ultra: (Brookfield, Middleboro, MA, USA).

Response 7: Correction has been made in the revised manuscript. (line 97, with highlight in red)

Point 8: Line 105: Please correct that the  UN110 Memmert was produced by a German company.

Response 8: Correction has been made in the revised manuscript. (line 95, with highlight in red)

Point 9: Line 107: The composite films were conditioned at room temperature but at what RH?

Response 9: Correction has been made in the revised manuscript. (line 109, with highlight in red)

Point 10: Line 110: According to ISO 37: 2017 dumb-bell spell with hyphens.

Response 10: Correction has been made in the revised manuscript. (line 112, with highlight in red)

Point 11: Line 118: Correct - tan δ.

Response 11: Correction has been made in the revised manuscript. (line 120, with highlight in red)

Point 12: Line 150: “Figure 1” – missing space.

Response 12: Correction has been made in the revised manuscript. (line 152, with highlight in red)

Point 13: Line 229: “…significantly increased…” Did you make any statistical analysis of the obtained results?

Response 13:  This has been changed to “increased” (line 256 with highlight in red)

Point 14: Line 271/ 368: Figure 6 and Figure 8. Please consider adding the 0,5 and 1,0 value (point) at the x-axis, same time remove 6.

Response 14: We agree with the reviewer’s comment but the we cannot do that with excel.

Point 15: Line 365: Correct - tan δmax.

Response 15: Correction has been made in the revised manuscript. (line 382, with highlight in red)

Point 16: References: Commas, not semicolons, in between the authors' names.

Response 16: Correction has been made in the revised manuscript. (References)

Reviewer 3 Report

The paper is well written and very precise. However some points could be face:

Results: Why only 300% modulus and  elongation at break were studied? Is it possible to study tensile strenght?

Antimicrobial activity: Do you have a picture of the biocomposite films that exert the antimicrobial activity?

In the introduction the authors can mention the property of chitosan for example: . Porta, L. Mariniello, P. Di Pierro, A. Sorrentino, C.V.L. Giosafatto. Transglutaminase crosslinked pectin- and chitosan-based edible films: a review. Critical Reviews in Food Science and Nutrition. 51: 223-238, 2011

Author Response

Response to Reviewer 3 Comments

Point 1: Results: Why only 300% modulus and  elongation at break were studied? Is it possible to study tensile strength?

Response 1: Yes, it is possible to study tensile strength of the composites. However, as the tensile strength tended to decrease with CT loading, we therefore selected only 300% modulus and  elongation at break to write the manuscript.

Point 2: Antimicrobial activity: Do you have a picture of the biocomposite films that exert the antimicrobial activity?

Response 2:  The pictures of clear zone of the biocomposite films have been incorporated in the revised manuscript. (Page 9, Figure 8)

Point 3: In the introduction the authors can mention the property of chitosan for example: . Porta, L. Mariniello, P. Di Pierro, A. Sorrentino, C.V.L. Giosafatto. Transglutaminase crosslinked pectin- and chitosan-based edible films: a review. Critical Reviews in Food Science and Nutrition. 51: 223-238, 2011

Response 3: The suggested reference has been incorporated into the revised manuscript. (reference no. 15)
